# Chitosan Oligosaccharide Lactate Increases Productivity and Quality of Baby Leaf Red Perilla

**Piotr Salachna** [1,*] and **Łukasz Łopusiewicz** [2,*]

1  Department of Horticulture, West Pomeranian University of Technology in Szczecin, 71-459 Szczecin, Poland
2  Center of Bioimmobilisation and Innovative Packaging Materials, West Pomeranian University of Technology in Szczecin, 71-270 Szczecin, Poland
*  Correspondence: piotr.salachna@zut.edu.pl (P.S.); lukasz.lopusiewicz@zut.edu.pl (Ł.Ł.);
   Tel.: +48-91-449-6359 (P.S.); +48-91-449-6135 (Ł.Ł.)

**Abstract:** Perilla (*Perilla frutescens*) belongs to the Lamiaceae family, is used as a spicy culinary herb leafy vegetable as well as medicinal and ornamental plant. However, little is known about protocols for baby leaf perilla production. Native chitosan is a well-known biostimulant used in crop plant production. Nevertheless, the influence of water-soluble chitosan oligosaccharide lactate (ChOL) on plant growth and bioactive compounds content remains unknown. The present pot experiment determines the effects of ChOL (0, 50 and 100 mg/L) on growth and selected biochemical characteristics of baby leaf red perilla. Compared to the untreated plants, ChOL application at 50 and 100 mg/L increased plant height (by 14.6% and 13.2%), the fresh weight of the above-ground part of plants (by 17.1% and 26.7%), leaves (by 21.8% and 35.5%) and roots (by 52.2%). The levels of total reducing sugars, polyphenolics, flavonoids and anthocyanins in perilla leaves were significantly higher in all plants treated with ChOL at all tested concentrations. This was confirmed by macromolecules (FT-IR) studies showing higher band intensity for key functional groups in leaf samples. The application of ChOL also enhanced the antioxidant activity by using DPPH, ABTS and $O_2^-$ radical scavenging activity assays. Based on the research, results suggested that ChOL may be used an effective plant biostimulant for high quality production of baby leaf red perilla.

**Keywords:** functional food; growth; oligo-chitosan; secondary metabolites; phytochemicals

## 1. Introduction

Recently, there has been an increased interest in the consumption of plants with a high content of bioactive substances [1]. Vegetables and herbs offered at early stages of their intensive growth, commonly known as baby leaves, are becoming especially popular [2]. The young leaves are soft, crisp and juicy, and therefore exceptionally tasty. In addition, the leaves at the initial stages of development contain elevated amounts of nutrients and health-promoting compounds [3]. Perilla (*Perilla frutescens* (L.) Britt., Lamiaceae) is a spice, oil-rich, aromatic and ornamental species belonging to the group of baby leaf plants [4]. High content of biologically active compounds makes perilla also useful in the treatment of many diseases such as: asthma, atopic dermatitis, diabetes, depression, hay fever, and metabolic syndrome [5]. From the industrial perspective, the most important species are *P. frutescens* var. *frutescens* grown for its oil and *P. frutescens* var. *crispa* grown for edible leaves with medicinal properties. Both varieties contain numerous chemotypes differing in the color and shape of their leaves [4]. Perilla is a rich source of compounds with antioxidant activity [6,7]. Its extracts show strong antiallergic, anti-inflammatory and spasmolytic effects, and they effectively lower the level of blood sugar, cholesterol, and triglycerides [4,5]. For these reasons perilla is classified as functional food [8]. To increase the productivity and quality of perilla crop, new strategies are being developed for its cultivation [9], including the use of compounds with biostimulating properties [10].

The main task of biostimulants is to support and stimulate plant growth and physiological processes [11]. At low concentrations the biostimulants, similarly to growth regulators, do not provide plants with nutrients, but they directly and indirectly affect plant growth and development [12,13]. One of such growth-promoting substances is chitosan, and its derivatives [14]. Chitosan is a highly popular biopolymer, composed of β(1→4)-D-glucosamine and *N*-acetyl-D-glucosamine subunits (Figure 1A), used in many branches of industry, medicine, pharmacy, and agriculture [14,15]. It is obtained from chitin, commonly present in exoskeletons of crustaceans and insects, as well as in fungal cell walls [16]. Both chitosan and its derivatives can exert a biostimulating effect on plants [17,18], and thus improve their growth and yield [19,20]. Chitosan stimulates plant growth and development by through direct and indirect regulating such as: cell division, cell elongation, macro and micronutrient uptake, chlorophyll synthesis, modulation of carbon and nitrogen metabolism, phytohormone activities and gene expression [21–24]. Plant treatment with chitosan may affect the level of biomacromolecules, including secondary metabolites such as polyphenolics and flavonoids, important for plant adaptation to the environmental conditions [25–27].

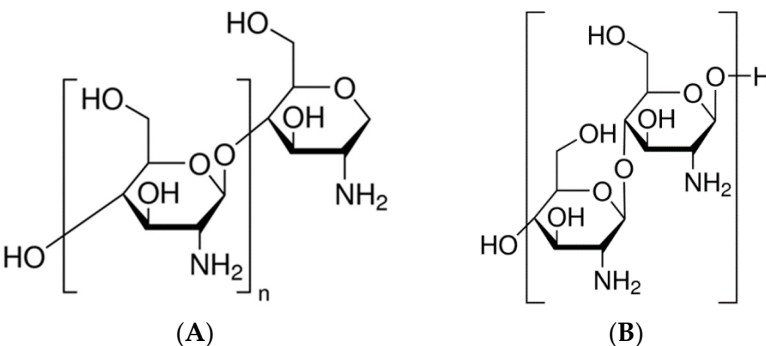

**Figure 1.** The chemical structure of (**A**) chitosan and (**B**) chitosan oligosaccharide lactate.

Chitosan covers a wide group of polycationic macromolecular compounds differing mainly in their molecular weight and degree of polydispersity. These parameters clearly influence most of the biological properties and functions of chitosan [28]. The factor limiting the use of chitosan is its poor solubility in a neutral environment. Chemical modifications of chitosan yield its derivatives with a much wider range of applications [29]. An example includes water-soluble oligomers of chitosan lactate (Figure 1B) with short chains and strictly defined molecular weight [30]. Degradation products of chitosan with low molecular mass may exhibit a better ability to promote plant growth and development than the original compound [31]. The effectiveness of chitosan-based biostimulants is highly diverse, which is why each product of this type requires an individual assessment. The precise mechanism of action of chitosan derivatives, especially its oligomers used as plant stimulants and elicitors, is still under investigation.

Oligosaccharide chitosan lactate (ChOL) is a product of chitosan degradation. It has low molecular mass, and a potential application as biostimulant for crop production. However, the effects of ChOL on the plant biomass yield and accumulation of secondary metabolites are unknown. Moreover, studies on application of chitosan or chitooligosaccharides for improving growth and yield of perilla have not previously been investigated. Therefore, in the present study, the effect of ChOL on growth, selected biochemical characteristics and antioxidant properties of baby leaf red perilla was examined. The impact of ChOL on the macromolecular composition of plant tissue using Fourier transform infrared spectroscopy (FT-IR) was also evaluated. We hypothesized that CHOL has a multi-directional, positive effect on plant growth and the content of bioactive metabolites and can be used as a potential biostimulant in baby leaf red perilla production.

## 2. Materials and Methods

### 2.1. Plant Material

Seeds of red perilla (*P. frutescens* var. *frutescens* f. *purpurea*) were purchased from the W. Legutko Breeding and Seed Company, Jutrosin, Poland and stored in paper bags in darkness at 20 °C.

### 2.2. Source of ChOL

The ChOL $(C_{12}H_{24}N_2O_9)_n$ with average molecular mass $(M_n)$ of 5000 g/mol, and the degree of deacetylation (DD) > 90% was purchased from Sigma Chemical Co., St. Louis, MO, USA.

### 2.3. Growth Conditions and Experimental Setup

Red perilla seeds were sown at the beginning of April into boxes filled with substrate TS 1 (Klasmann-Deilmann, Warsaw, Poland) with pH 5.5. The boxes were placed in a greenhouse belonging to the Department of Horticulture, West Pomeranian University of Technology in Szczecin (53°25′ N, 14°32′ E) with day/night temperature set at 22/18 °C. Three-week-old seedlings of similar size were transplanted individually into plastic pots (8 cm height and 11 cm diameter) filled with deacidified peat with pH 5.7 amended with 12N-4.8P-15K mineral fertilizer (Yara International ASA, Oslo, Norway) used at 1.5 g/L. Each pot was placed in a 500 mL beaker (Figure 2A,B). The plants grown in a greenhouse under natural photoperiod were drenched after potting four times, at weekly intervals, with 50 and 100 mg/L ChOL solutions. The concentration of ChOL was determined based on the results of a pilot study. Each time 50 mL of the solution per pot was applied. Control plants were drenched with tap water. The experimental design was set in randomized complete block design and each treatment was replicated four times with ten plants per experimental unit.

### 2.4. Growth Traits

All plants were harvested after eight weeks after transplanting (Figure 2A) and the plant height (from the substrate level to the top of the plant), fresh weight of above-ground part, leaves and roots per plant were determined. The leaves collected from upper external parts of the plants were dried up in darkness for three weeks at 25 °C and powdered.

### 2.5. Determination of Reducing Sugars and of Free Amino Acids Content

Reducing sugars and of free amino acids content analyses were performed according to our previous report using the DNS method (3,5-dinitrosalicylic acid) and ninhydrin-Cd reagent [32].

### 2.6. Determination of Total Anthocyanins Content

The anthocyanins were extracted with 50 mL of methanol acidified with 0.5% acetic acid and estimated based on a previously reported method [7]. Extractions and analyzes were carried out in triplicate. Total anthocyanin content was expressed as milligrams of cyanidin-3-glucoside equivalent per gram of dry weigh.

### 2.7. Determination of Total Phenolic Content and Total Flavonoids Content

One gram of dried samples were mixed with 50 mL of methanol/water solution (7:3 *v/v*), and subsequently extracted in an ultrasonic bath (Elmasonic S30H, Elma Schmidbauer GmbH, Singen, Germany) for 15 min. The obtained extracts were centrifuged at 14,000 rpm for 5 min at 20 °C (Centrifuge 5418 Eppendorf, Warsaw, Poland), then filtered using 0.22 μm nylon membrane filters (Sigma-Aldrich, Darmstadt, Germany). The resulting clear extracts were used for further analyses. The total phenolic content (TPC) and total flavonoid content (TFC) were assayed using spectrophotometric methods following the methodology described elsewhere [32]. Briefly, 20 μL of the extracts was mixed with 150 μL of distilled water and 100 μL of Folin–Ciocalteu reagent, then incubated 5 min. Subsequently, the

mixture was mixed with 80 μL of Na$_2$CO$_3$ (7.5%), and then incubated for 30 min at 40 °C without light. Absorbance was measured at 765 nm using a microplate reader (Synergy LX, BioTek, Winooski, VT, USA). Gallic acid (GAE) was used to prepare the standard curve and the results were expressed as milligrams GAE per g dry weight of plants (mg GAE/g DW). To determine TFC of the samples, 250 μL of the extracts was mixed with 100 μL of distilled water and 7.5 μL of NaNO$_2$. The mixture was supplemented with 7.5 μL of 10% AlCl$_3$ solution after 5 min of incubation. The samples were left for 6 min at room temperature, then 25 μL of 1 M NaOH were added, and the mixture was diluted with 135 μL of distilled water. Absorbance was measured at 510 nm using microplate reader. Quercetin served for the preparation of the calibration curve and the results were expressed as mg quercetin equivalent (QE) per g dry weight of plants (mg QE/g DW).

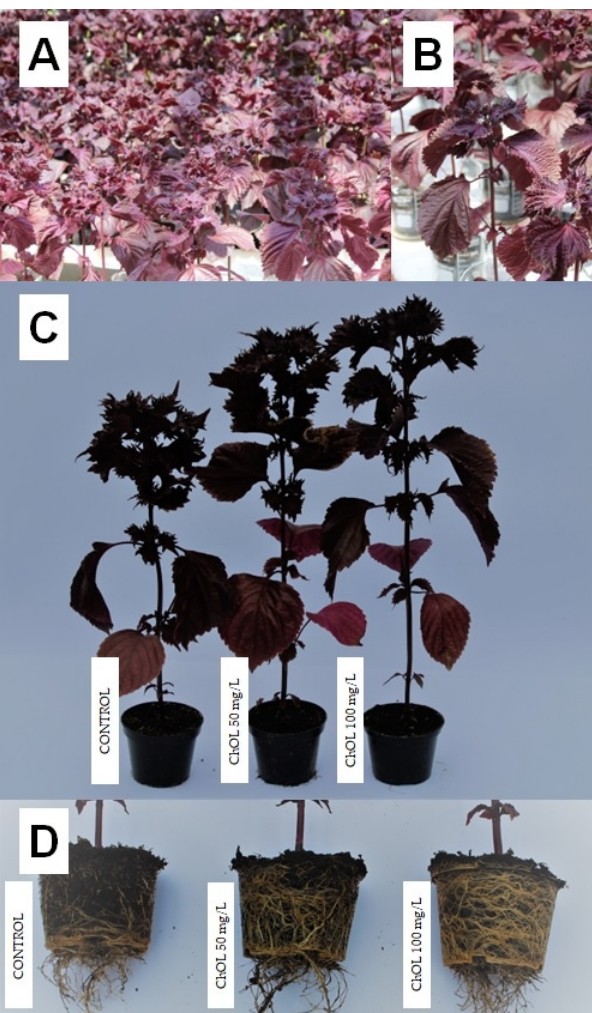

**Figure 2.** Effects of chitosan oligosaccharide lactate (ChOL) on plant growth of red perilla: (**A**,**B**) photograph of plants eight weeks after transplanting; (**C**) photograph of whole plants and (**D**) root system (left to right: nontreated control, 50 and 100 mg/L ChOL).

## 2.8. Determination of DPPH, ABTS and Superoxide (O$_2^-$) Scavenging Activity

For the antioxidant properties of red perilla leaves DPPH, ABTS and O$_2^-$ radicals scavenging activities tests were chosen and the assays were performed based on methodology described previously [33]. Briefly, DPPH radical scavenging activity was determined by mixing 1 mL of extracts with 1 mL of 0.01 mM DPPH methanolic solution. Absorbance was determined at 517 nm. Three mL of ABTS$^{·+}$ solution were mixed with 50 μL of extracts and absorbance was measured at 734 nm. Three mL of 50 mmol/L (pH 8.2) Tris-HCl buffer were mixed with 1 mL of the extracts. These mixtures were mixed with a pyrogallol solution

(0.3 mL, 7 mmol/L, preheated to 25 °C) and allowed to react for exactly 4 min, then 1 mL od 10 mmol/L of HCl was added to terminate the reaction, and absorbance was measured at 318 nm.

### 2.9. Fourier-Transform Infrared Spectroscopy (FT-IR) Studies

FT-IR analyses were performed using a FT-IR spectrometer (Perkin Elmer 100 Spectrometer, Waltham, MA, USA). The samples of leaves of treated and untreated plants were collected from five plants and scanned in the wavenumber range 700 cm$^{-1}$–4000 cm$^{-1}$ (100 scans, resolution 4 cm$^{-1}$). The resulting FT-IR spectra were normalized, corrected against the baseline and analyzed using SPECTRUM software (v10, Perkin Elmer, Waltham, MA, USA).

### 2.10. Statistical Analysis

Data were analyzed using the Statistica 13.3 (TIBCO Software Inc. Statsoft, Cracow, Poland) software by performing a one way analysis of variance (ANOVA). Confidence half-intervals were calculated based on Tukey's HSD test at the 95% confidence limits. Pearson's linear correlation coefficient between total anthocyanins, polyphenols, flavonoids content and antioxidant activity was also calculated. All data was presented as means plus or minus standard errors.

### 3. Results and Discussion

The data in Table 1 and Figure 2C,D demonstrate that plant treatment with 50 and 100 mg/L ChOL resulted in a significant improvement in plant height (14.6% and 13.2%), fresh weight of the above-ground parts (17.1% and 26.7%), fresh weight of the leaves (21.8% and 35.5%) and fresh weight of roots (52.2%) versus control plants treated with water. The weight of the above-ground part was more strongly stimulated by the higher ChOL dose. A stimulating effect of chitosan degradation products on plant height was reported in *Solanum tuberosum* [17] and *Triticum aestivum* [34], while biomass increase following application of chitosan derivatives was noticed in *Cordyline terminalis* [35], *Oryza sativa* [36] seedlings, and *Mentha piperata* [23]. As shown in Figure 3A, application of ChOL with 50 and 100 mg/L significantly increased reducing sugars content in leaves of red perilla, compared to the control by 2.36- and 2.41-fold, respectively. Plants treatment with ChOL did not affect total free amino acids content (Figure 3B). Similar to this study, Zhang et al. [37] reported that some chitooligomers fractions with low molecular mass could increase the soluble sugar content of *T. aestivum* seedlings. Chitosan and its oligomers support development of the rhizosphere biota and root system, effective uptake of nutrients and water by the roots, and intensify physiological processes, including $CO_2$ assimilation and accumulation of sucrose content [11,22,24]. All these produce the final result of stimulated growth and increased plant biomass [19,20].

**Table 1.** Effects of chitosan oligosaccharide lactate (ChOL) on plant height, fresh weight of above-ground part, leaves and roots per plant of red perilla.

| Treatment | Plant Height (cm) | Fresh Weight of Above-Ground Part (g/plant) | Fresh Weight of Leaves (g/plant) | Fresh Weight of Roots (g/plant) |
|---|---|---|---|---|
| Control | 42.5 ± 1.3 [b1] | 17.6 ± 0.3 [c] | 11.0 ± 0.3 [b] | 3.83 ± 0.19 [b] |
| ChOL 50 mg/L | 48.7 ± 0.3 [a] | 20.6 ± 0.2 [b] | 13.4 ± 0.5 [a] | 5.83 ± 0.19 [a] |
| ChOL 100 mg/L | 48.1 ± 0.6 [a] | 22.3 ± 0.5 [a] | 14.9 ± 0.5 [a] | 5.83 ± 0.23 [a] |
| Significance | 0.00022 | 0.00017 | 0.00271 | 0.00003 |

[1] Means over each column not marked with the same letter are significantly different at $p \leq 0.05$. Data are expressed as mean and standard errors of the mean (±SEM).

Plants are a rich source of biologically active secondary metabolites, often with antioxidant properties that inhibit oxidation reactions in the body [38]. The antioxidant activity is particularly strong in polyphenolic compounds (especially flavonoids), with well-known

health-promoting features [39,40]. Elicitation is an efficient method used for boosting the production of plant secondary metabolites [41]. Chitosan oligomers, despite their considerable potential [42], are still relatively little known elicitors. Our study investigated if oligochitosan lactate affects the metabolite level and antioxidant activity of red perilla leaf extracts. As shown in Figure 4A–C, the treatment with ChOL at 50 and 100 mg/L significantly enhanced total content of total anthocyanins (by 38.2% and 29.7%), polyphenols (by 11.2% and 18.1%) and flavonoids (by 51.7% and 59.5%), as compared with the control plants. ChOL concentration did not seem to affect their content. In *Melisa officinalis*, chitosan lactate at 100 and 500 mg/L enhanced the accumulation of anthocyanins [26]. Growing accumulation of total phenols with rising concentration of oligochitosan (25–100 mg/L) was reported in *C. terminalis* [35], and a stimulating effect of chitosan oligosaccharides at 50 and 100 mg/L on the production of total flavonoids and total polyphenols was detected in *Origanum vulgare* ssp. *hirtum* [43]. It is hypothesized that chitosan and its low molecular weight oligomers can upregulate the expression levels of genes involved in metabolite biosynthesis and induce the activity of plant defence enzymes, resulting in increased accumulation of secondary metabolites, such as polyphenols, lignin, flavonoids, and phytoalexins [43–45].

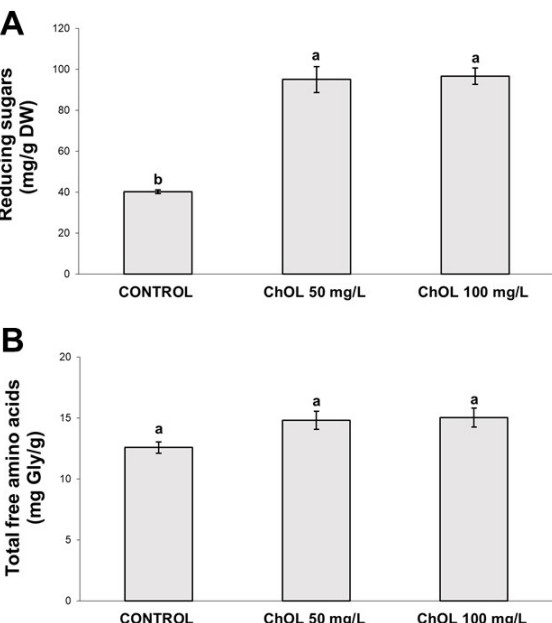

**Figure 3.** Effects of chitosan oligosaccharide lactate (ChOL) on reducing sugars (**A**) and total free amino acids (**B**) content in leaves of red perilla. Bars represent the average and error bars show standard errors of the mean. Different letters above and below bars indicate significant differences at $p \leq 0.05$.

Antioxidant activity is the ability of a substance to react with oxidative compounds, that is with molecular and radicals. Figure 4D–F shows the antioxidant properties of the investigated red perilla extracts. The plants treated with 50 and 100 mg/L ChOL were significantly more efficient at neutralizing free DPPH (by 17.0% and 17.1%) and ABTS (by 29.8% and 32.5%) radicals than the control plants. Similarly higher (by 6.1% and 7.2%) superoxide ($O_2{}^-$) scavenging activity was observed for the extracts derived from plants treated with 50 and 100 mg/L ChOL. The antioxidant activity of red perilla assessed by three different methods did not depend on ChOL concentration. Increased radical scavenging activity following the application of some chitosan derivatives was also reported in the seeds of *Glycine max* during germination [31]. Chitosan derivatives may play an important role in removing reactive oxygen species by inducing the activity of the plant antioxidant system [7,8,45]. The elevated antioxidant potential of the extracts from plants treated with ChOL may result from their increased content of secondary metabolites. This

was confirmed in our study by a good correlation between total anthocyanins, polyphenol, as well as flavonoids content and antioxidant activity assessed by methods based on DPPH (r = 0.965, r = 0.878 and r = 0.729, respectively), ABTS (r = 0.867, r = 0.856 and r = 0.623, respectively), and superoxide ($O_2^-$) scavenging activity (r = 0.819, r = 0.921 and r = 0.679, respectively). Anthocyanins, polyphenolics and flavonoids are known and crucial players in the process of free radical scavenging, which is reflected by a positive correlation between their content and antioxidant properties of many plant materials [46].

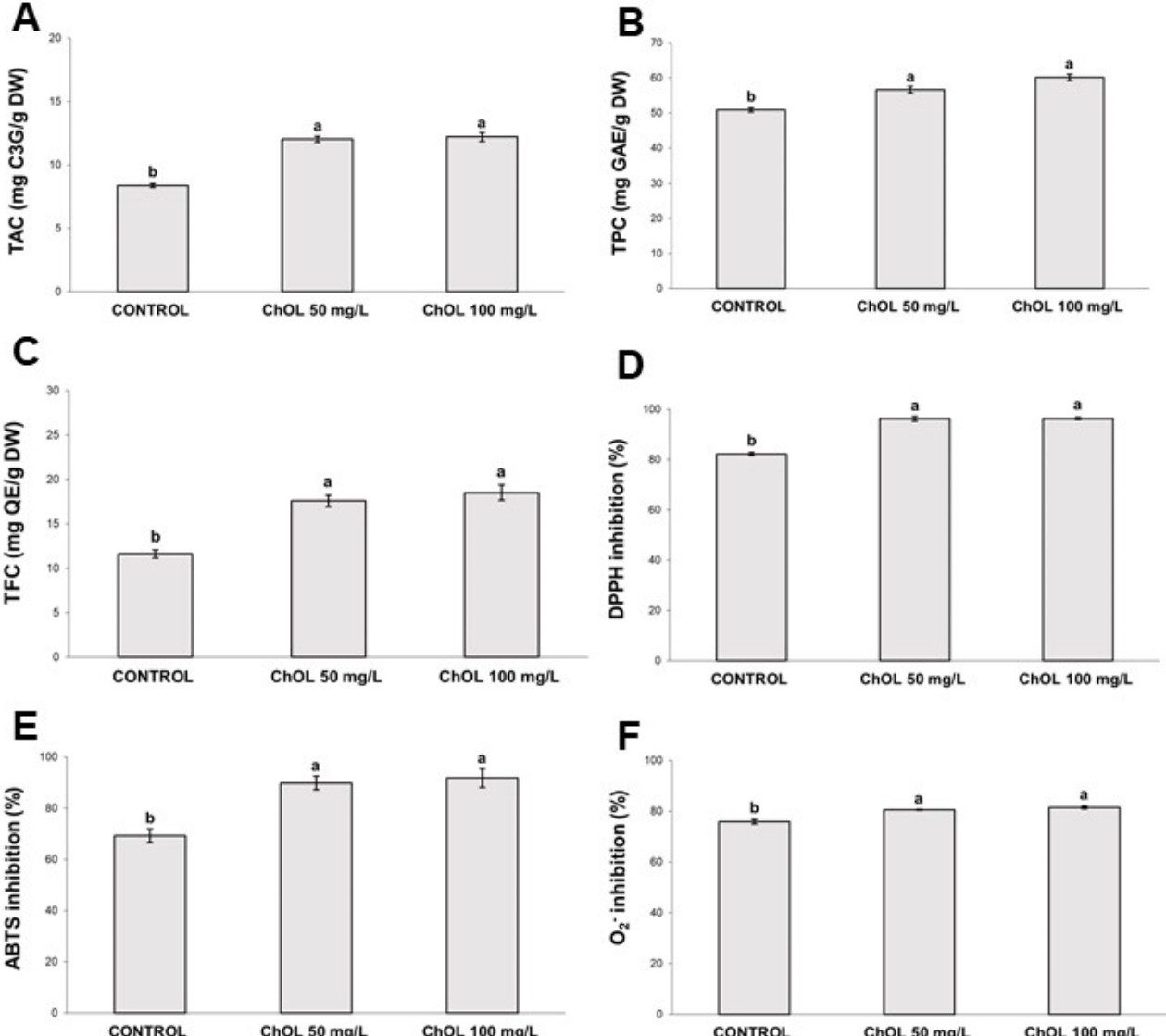

**Figure 4.** Effects of chitosan oligosaccharide lactate (ChOL) on (**A**) total anthocyanin content (TAC); (**B**) total phenolic content (TPC); (**C**) total flavonoid content (TFC) and antioxidant activities: (**D**) DPPH, (**E**) ABTS and (**F**) $O_2^-$ inhibition of red perilla. Bars represent the average and error bars show standard errors of the mean. Different letters above and below bars indicate significant differences at $p \leq 0.05$. C3G: cyanidin-3-glucoside equivalent; GAE: gallic acid equivalent; QE: quercetin equivalent; DW: dry weight; DPPH: (1,1-diphenyl-2-picryl-hydrazyl); ABTS: 2.2′-azinobis(3-ethylbenzothiazoline-6-sulfonic acid) diammonium salt.

FT-IR spectroscopy can provide comprehensive data on macromolecule composition in plants [47]. This method allows for identification of individual functional groups of macromolecules in the tested material and for tracking the changes occurring in various chemical bonds. As can be seen in Figure 5, the analysis of leaf spectra for red perilla revealed that

ChOL increased the intensity of bands centered around 3280 cm$^{-1}$, 2919 cm$^{-1}$, 2850 cm$^{-1}$, 1730 cm$^{-1}$, 1602 cm$^{-1}$, 1415 cm$^{-1}$, 1149 cm$^{-1}$ typical of proteins, lipids, and carbohydrates (–OH, –NH$_2$, –CH, –CH$_3$, –CH$_2$, O=C and C=C groups) [47,48]. Differences were particularly detected for the ChOL-treated samples in the region 1415-1149 cm$^{-1}$ where CH$_2$ wagging vibrations occur that may correspond to aromatic and ring vibrations, followed by changes in the amide, phenolic and aromatic band contributions (1700–1500 cm$^{-1}$), where compounds such as lignin, tannin or anthocyanins (that are mainly aromatic) could be contributing [49]. Moreover, greater intensity of bands denoting –OH groups (approximately at 3280 cm$^{-1}$) in plants grown in the presence of ChOL may be due to greater content of polyphenols and greater antioxidant activity discussed above. Compounds with antioxidant properties contain –OH group that reduces free radicals, and polyphenols are molecules that may contain two, three or more hydroxyl groups on their aromatic ring [40]. However, to better understand the effects of ChOL on plant composition and structure it is necessary to reinforce the results of macromolecules FT-IR analyses obtained by the spectroscopic method by further in-depth biochemical and molecular studies.

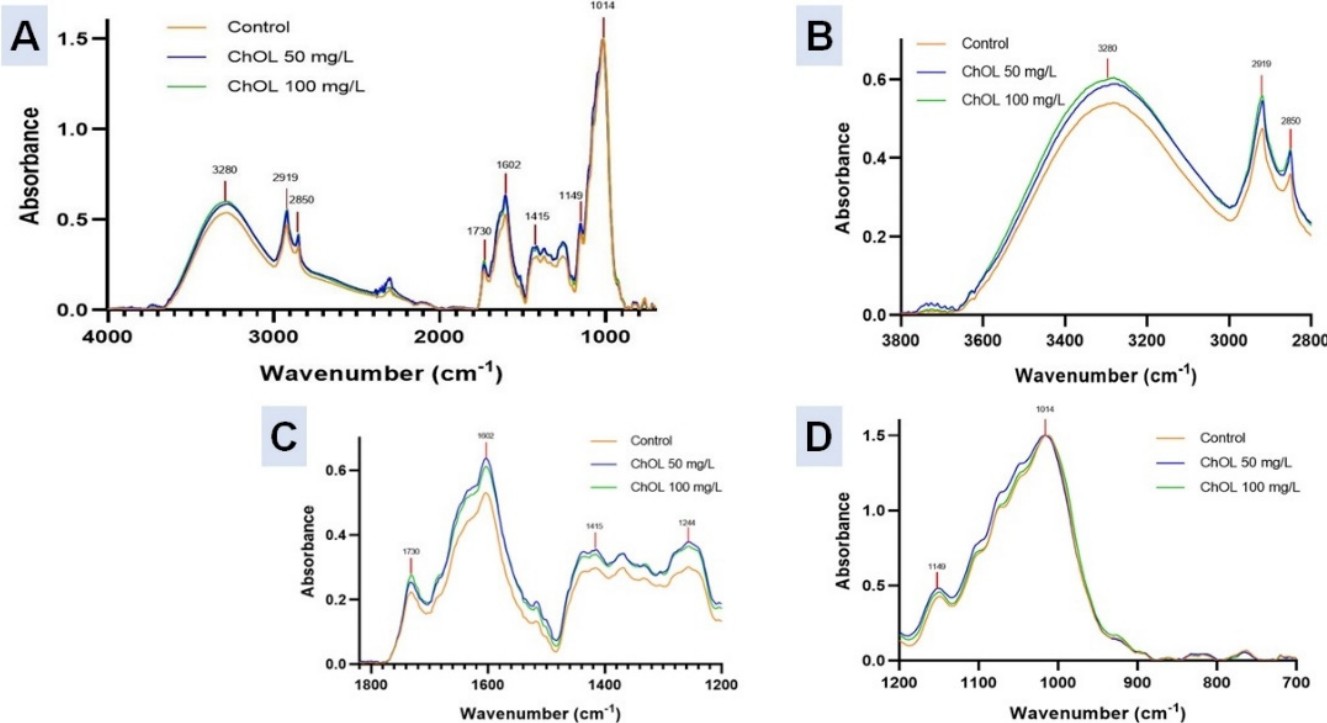

**Figure 5.** Fourier transform infra-red spectroscopy (FT-IR) spectra of red perilla treated with chitosan oligosaccharide lactate (ChOL) (**A**) whole FT-IR spectrum (4000–700 cm$^{-1}$); (**B**) in the range of 3600–2400 cm$^{-1}$ (–OH, –NH, –CH$_2$ and –CH$_3$ groups); (**C**) in the range of 1800–1200 cm$^{-1}$ (proteins); (**D**) in the range of 1250–700 cm$^{-1}$ (polysaccharides).

## 4. Conclusions

Chitosan in the form of ChOL is a natural, safe and environmentally friendly product, showing the features of a biostimulator and an elicitor and meeting the principles of sustainable agriculture. ChOL first applied in the production of red perilla baby leaf effectively improved plant productivity manifested in their increased biomass of the above-ground part of plants and root system. Moreover, the application of ChOL enhanced the health-promoting properties of the leaves, which were abundant in anthocyanins and total polyphenols and flavonoids, and showed higher antioxidant activity. Further research should focus on the effects of ChOL on targeted key metabolites, their profile and bioactivity of the investigated baby leaf red perilla.

**Author Contributions:** Conceptualization, P.S. and Ł.Ł.; methodology, P.S. and Ł.Ł.; data analysis, P.S. and Ł.Ł.; writing—original draft preparation, P.S. and Ł.Ł.; writing—review and editing, P.S. and Ł.Ł. All authors have read and agreed to the published version of the manuscript.

**Funding:** Presented here research received external funding in project "ZUT 2.0—Modern Integrated University" grant number POWR.03.05.00-00-Z205/17. This research was funded by the National Science Centre, Poland (Project 2018/02/X/NZ9/03281).

**Data Availability Statement:** The data presented in this study are available within the article.

**Conflicts of Interest:** The authors declare no conflict of interest.

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
