# Peer review of "Chitosan Oligosaccharide Lactate Increases Productivity and Quality of Baby Leaf Red Perilla"

_agronomy, doi:10.3390/agronomy12051182_

Round 1

Reviewer 1 Report

The work deals about the effect of chitosan oligosaccharide lactate (ChOL) on productivity and quality of Baby Leaf Red Perilla.

The results obtained suggest that ChOL may be used as plant bio stimulant for increase of quality of baby leaf red perilla.

We think that the manuscript needs to be improved, following suggestions below.

-Although the authors underline that the spectroscopic method should be reinforced by further biochemical and molecular analysis, I think that is necessary to implement the data obtained by the FT-IR.

-The English language must be improved.

-The values reported in manuscript (lines 184-187) do not correspond to values reported in the Table1.

-The lines 184 reports the figure 2, but it is figure 3 (A, B). In all manuscript the number of figures is wrong, also in the captures.

Author Response

The work deals about the effect of chitosan oligosaccharide lactate (ChOL) on productivity and quality of Baby Leaf Red Perilla. The results obtained suggest that ChOL may be used as plant bio stimulant for increase of quality of baby leaf red perilla. We think that the manuscript needs to be improved, following suggestions below.

Answer: Thank you very much for your time spent on a careful and detailed revision of our manuscript. Below you will find our answers to all the remarks. We hope our explanations are comprehensive and will dispel any possible doubts.

-Although the authors underline that the spectroscopic method should be reinforced by further biochemical and molecular analysis, I think that is necessary to implement the data obtained by the FT-IR.

Answer: We agree with this reviewer's comment and have therefore modified the last sentence of the Results and Discussion section.

-The English language must be improved.

Answer: Linguistic errors were corrected.

-The values reported in manuscript (lines 184-187) do not correspond to values reported in the Table1.

Answer: We apologize for this mistake. The values was revised.

-The lines 184 reports the figure 2, but it is figure 3 (A, B). In all manuscript the number of figures is wrong, also in the captures.

Answer: We apologize for this mistake. Figure numbers and captions are corrected.

Reviewer 2 Report

Review Remarks on the manuscript [Chitosan Oligosaccharide Lactate Increases Productivity and Quality of Baby Leaf Red Perilla]. Overall, the presentation of the manuscript is good. However, there are some general points and scientific queries below-mentioned needed to be addressed in the manuscript:

  • What were the criteria for the selection of ChOL’s concentration? Was it literature or some preliminary experiments? Please check and include.
  • In line 39: many diseases. Give some examples.
  • In line 182: means plus or minus standard error. I think errors, please check, and replace.
  • In Figure 2: if the authors could mention the doses of ChOL in the C and D parts, it will be great.
  • The FTIR spectra are unable to depict the variations in wavenumber with the treatment of ChOL in comparison to control. I know the pattern is almost similar but the authors stated that there were alterations in the region 1415-1149 cm-1. So, it would be great if authors could zoom out this region or provide good quality spectra to demark the differentiation.
  • Pearson’s correlation matrix can be included in the revised version.
  • The line 303-304 in the conclusion failed to reflect their purpose; please include appropriate concluding remarks because the study is mainly focused on the profiling phytochemical, antioxidants, and growth enhancement.

Author Response

Answer: Thank you very much for your time spent on a careful and detailed revision of our manuscript. Below you will find our answers to all the remarks. We hope our explanations are comprehensive and will dispel any possible doubts.

What were the criteria for the selection of ChOL’s concentration? Was it literature or some preliminary experiments? Please check and include.

Answer: We added this information.

In line 39: many diseases. Give some examples.

Answer: This sentence was improved.

In line 182: means plus or minus standard error. I think errors, please check, and replace.

Answer: Changed as suggested.

In Figure 2: if the authors could mention the doses of ChOL in the C and D parts, it will be great.

Answer: This figure was improved.

The FTIR spectra are unable to depict the variations in wavenumber with the treatment of ChOL in comparison to control. I know the pattern is almost similar but the authors stated that there were alterations in the region 1415-1149 cm-1. So, it would be great if authors could zoom out this region or provide good quality spectra to demark the differentiation.

Answer: We would like to thank you for these suggestions. This figure was improved.

Pearson’s correlation matrix can be included in the revised version.

Answer: Values of the correlation coefficient are included in the paper and therefore we prefer not to repeat them in the correlation matrix.

The line 303-304 in the conclusion failed to reflect their purpose; please include appropriate concluding remarks because the study is mainly focused on the profiling phytochemical, antioxidants, and growth enhancement.

Answer: We would like to thank you for these suggestions. The last sentence in the conclusion was changed as recommended.